# Characterization and Immunomodulatory Effects of High Molecular Weight Fucoidan Fraction from the Sporophyll of *Undaria pinnatifida* in Cyclophosphamide-Induced Immunosuppressed Mice

**DOI:** 10.3390/md17080447

**Published:** 2019-07-29

**Authors:** Hee Joon Yoo, Dong-Ju You, Kwang-Won Lee

**Affiliations:** 1Department of Biotechnology, College of Life Science and Biotechnology, Korea University, Seoul 02841, Korea; 2Haerim Fucoidan, Wando-gun, Jeollanam-do 59108, Korea

**Keywords:** fucoidan, NK activity, immunosuppression, high molecular weight fucoidan

## Abstract

Immunomodulation involves two mechanisms, immunostimulation and immunosuppression. It is a complex mechanism that regulates the pathophysiology and pathogenesis of various diseases affecting the immune system. Immunomodulators can be used as immunostimulators to reduce the side effects of drugs that induce immunosuppression. In this study, we characterized the chemical composition of high molecular weight fucoidan (HMWF) and low molecular weight fucoidan and compared their functions as natural killer (NK) cell-derived immunostimulators in vitro. We also tested the effectiveness of HMWF, which has a relatively high function in vitro, as an immunostimulator in immunosuppressed animal models. In these models, HWMF significantly restored NK cell cytotoxicity and granzyme B release to the control group level. In addition, the expression of interleukin (IL)-1β, IL-2, IL-4, IL-5, IL-12, interferon (IFN)-γ, and tumor necrosis factor (TNF)-α also increased in the spleen. This study suggests that HMWF acts as an effective immunostimulant under immunosuppressive conditions.

## 1. Introduction

There are two mechanisms for immunomodulation—immune stimulation and immunosuppression. It is a complicated process that controls the pathogenesis and pathophysiology of different illnesses influencing the immune system. Immunomodulators can be used as immunostimulants to decrease drug-inducing side effects, such as immunosuppression [1]. Cyclophosphamide (CY), one of the most commonly prescribed anticancer agents since 1958, has been used to treat autoimmune and immune-mediated diseases with prodrugs and immunosuppressive agents for cancer therapy [2]. This compound is an alkylating agent, and the main mechanism of action is DNA alkylation, which inhibits DNA synthesis and induces apoptosis [3]. However, CY also destroys normal cells, including immune cells, via the cytokine storm [4,5]. In fact, low CY concentrations selectively stimulate immune cells to enhance antitumor effects on existing tumor cells, while high CY concentrations are known to exhibit cytotoxicity as well as antitumor effects [6]. Therefore, when cancer therapy-based chemotherapy, such as CY is performed, it is important to prevent the adverse effects of cytotoxicity and homeostatic collapse caused by immune degradation. Immunomodulators can be used in these situations as immune stimulators to reduce the side effects of drugs that induce immunosuppression [1].

Fucoidan is a sulfated polysaccharide derived from algae [7] which consists of sulfate, uronic acid, and sugars, such as fucose, galactose, and maltose. It is known to have many biological activities and has been known to act as an anticoagulant and antiviral, and function in immunity enhancement, anti-inflammation, blood lipid reduction, antioxidation increased liver and kidney activity, and gastrointestinal protection [7,8,9,10]. Fucoidans differ widely in biological activity depending on species, seasonality, extraction methods, fractionation methods, and their molecular weight (MW) [11,12]. Recently, studies on the functional and immunomodulating ability of a fucoidan extracted from the sporophyll of *Undaria pinnatifida* have been reported [13,14,15,16,17,18,19]. However, the in vivo immunomodulating ability of low molecular weight fucoidan (LMWF) and high molecular weight fucoidan (HMWF) in NK cells and the immunoregulatory function in immunosuppression in vivo have not been studied.

Here, we attempt to characterize fucoidan by comparing the chemical components of LMWF and HMWF, and showing that HMWF has significant immunoregulatory and immune-promoting functions in an immunosuppressed animal model.

## 2. Results and Discussion

### 2.1. Chemical Composition of the HMWF and LMWF

The carbohydrate contents of HMWF and LMWF were 72.0% and 73.8%, respectively, and they both displayed similar monosaccharide composition. Their fucose contents were 21.0% and 21.5%, respectively. Previous studies have shown that the ratio of fucose to galactose in fucoidan extracted from homozygous sporophyll is close to 1:1 [10,12,20]. In the monosaccharide composition of HMWF and LMWF, only galactose contents were noticeably different. HMWF showed a similar ratio, but LMWF had a relatively higher fucose content than galactose. The composition of fucoidan is usually affected by factors such as harvesting time and extraction method [11,21]. Uronic acid contents of HMWF and LMWF were similar. Sulfate levels, known to be closely related to the bioactivity of fucoidan [21,22,23,24], were 30.9% and 28.8% for HMWF and LMWF, respectively (Table 1). Using analytical methods, the links between fucoidan monosaccharides were discovered to be 1-3 linked fucose and 1-3, 1-4, and 1-6 linked galactose in the glycosidic bonds, and sulfate substitution sites are primarily at C2- or C4-positions of residues of fucose, and C3- or C6-positions of residues of galactose [11]. The structure of HMWF and LMWF fractionated according to MW via cross-flow ultrafiltration from *U. pinnatifida* was not determined in our study. Given the distinction between HMWF and LMWF in the content of galactose residues and sulfate content, it can be seen that these two fucoidans appear to differ in structure. Future study on the structural characterization of HMWF and LMWF is required.

### 2.2. The molecular Weight of HMWF and LMWF

The MW of fucoidan has been considered as one of several factors that affect its functional properties [25,26]. The MWs of HMWF and LMWF fractions were determined via high-performance gel permeation chromatography (HPGPC) (Figure 1). Pullulan (Shodex, Showa Denko, Japan) was used as a standard and showed elution profiles similar to fucoidan fractions, displaying a wide range of MWs, from as low as 1.32 kDa to 1,330 kDa. The HMWF fraction showed a prominent peak at about 258.7 kDa, and the LMWF had a large peak at about 54.1 kDa.

### 2.3. Effect of HMWF and LMWF on the Proliferation of NK-92MI Cells

Figure 2 shows the degree of proliferation after treatment of NK-92MI cells, a human natural killer cell line, with HMWF and LMWF for 24 h. The degree of proliferation of the group treated with HMWF (62.5 to 2,000 μg/mL) was significantly increased in a dose-dependent manner. The proliferation of the group treated with 250 μg/mL of HMWF was comparable to the proliferation of concanavalin A (ConA, 0.5 μg/mL), which served as the positive control. ConA is a well-known lymphocyte mitogen that increases NK cell proliferation [27]. In addition, the group treated with HMWF at 2,000 μg/mL showed about 143% higher proliferation compared to the untreated control group. The degree of proliferation of the group treated with LMWF (500 to 1,000 μg/mL) increased, but not significantly when compared to the control. In the HMWF and LMWF groups treated with the same concentration (1,000 μg/mL), 131% and 108% proliferation were observed, respectively, compared to the control group. As a result, HMWF is a more potent agent of NK cell proliferation and functions more as an immune stimulator than LMWF. NK cells engage in target cell lysis and activate cytokine secretion [28] as well. Therefore, if treated at the same concentration, a marked rise in the number of NK cells in the HMWF-treated group means that it may be more effective in immunomodulation.

### 2.4. Effect of HMWF and LMWF on NK Cell Cytotoxic Activity

After NK-92MI cells were treated with HMWF and LMWF, NK cell cytotoxic activity against YAC-1 cells, as target cells was measured at three E:T ratios (2:1, 5:1, 12.5:1) (Figure 3). The cell cytotoxic activity of the control group was 63% at a 2:1 E:T ratio, while the cell cytotoxic activities of HMWF and LMWF treated cells were 76% and 70%, respectively. The cell cytotoxic activity of ConA (5 μg/mL) used as a positive control was 70%, similar to the cell cytotoxic activity of the HMWF-treated group. The cell cytotoxic activity of the untreated group at E:T ratio 5:1 was 73%, and for the groups treated with HMWF and LMWF were 78% and 75%, respectively. At E:T ratio 12.5, cell cytotoxic activity was 84% in the untreated group, and 86% and 84% in the HMWF and LMWF groups, respectively. As the E:T ratio increased, the cytotoxicity of NK cells in the target cells tended to increase slightly. And the cell cytotoxic activity increased effectively in the HMWF-treated group at E:T ratio 2:1. These results suggest that HMWF has a relatively higher immunoregulatory ability of NK cytotoxicity.

### 2.5. Effect of HMWF and LMWF on Splenic NK Cell Cytotoxic Activity and Granzyme B Release in CY-Treated Mice

The control group (CON) and CY-treated only group (CY only) were orally administered with vehicles alone for 28 d. The CY only group, HMWF-, and LMWF-treated groups were injected with CY on day 21. The NK cell cytotoxic activity of splenocytes extracted from the spleen of harvested mice and the production of granzyme B were measured at three E:T ratios (4:1, 10:1, and 25:1) for YAC-1, the target cells (Figure 4). As shown in Figure 4A, compared with the CON group at all E:T ratios, NK cell activity decreased in all CY-treated groups. In contrast, the splenic NK cell cytotoxic activity was significantly increased (*p* < 0.05) in the group with 25 or 75 mg of HMWF/kg body weight (b.w.), which was administered orally. At a 4:1 E:T ratio, NK activity in the CY only group was significantly reduced to 7%, while NK activity of HMWF- treated groups were significantly recovered by 16%. A similar tendency was observed at E:T ratios of 10:1 and 25:1. NK activity was recovered about 139% and 149% at an E:T ratio of 4:1 in HMWF high and HMWF low, respectively, when compared with the CY only group.

As shown in Figure 4B, the production of granzyme B reduced by CY was also significantly increased (*p* < 0.05) in the HMWF high group. Granzyme B is a pro-apoptotic protein that is known to initiate the death of target cells [29], and NK cell cytotoxicity is mediated by the expression of cytolytic granules containing granzyme B and perforin [30,31]. It can be interpreted that NK cells in the spleen activated by oral administration of HMWF increased granzyme B production to reduce the number of target YAC-1 cells.

In the above experiment, the splenic NK cell cytotoxic activity and granzyme B release recovered to near normal level in the group treated with HMWF orally, which showed a tendency to increase significantly at one or more E:T ratios compared to the CY only group. Thus, it implies that HMWF in the immunosuppressive model can function as an immunomodulator.

### 2.6. Effect of HMWF on Splenic Cytokine Gene Expression in the CY-Treated Mice

The spleen is a crucial lymphoid organ composed of immune cells, such as monocytes, macrophages, and B and T lymphocytes, that mediate both innate and acquired immunity [32]. Cytokines regulate both cellular and humoral immune responses by affecting immune cell proliferation, differentiation, and functions [33]. Thus, we examined the expression of cytokines at the mRNA level in the spleen (Figure 5) in order to confirm the efficacy of immunity. Interleukin (IL) -2 and interferon (IFN)-γ, which are type 1 T helper (Th-1)-derived cytokines, play important roles in cell-mediated immune response [34]. Th1 cells produce IFN-γ, IL-2, and IL-12, and promote macrophage activation and production of opsonizing and complement-fixing antibodies [35]. In addition, IL-4, IL-5, IL-6, and IL-10 are type 2 T helper (Th-2)-derived cytokines that promote humoral immunity [36]. Several other proteins are secreted by both Th-1 and Th-2 cells, including tumor necrosis factor-α (TNF-α) and granulocyte-macrophage-colony-stimulating factor, GM-CSF [37]. Recent studies have shown that CY inhibits both the humoral and cellular immune responses [38,39], suggesting that the level of each cytokine mRNA expression in all groups treated with CY alone is reduced by approximately 10% to 49%. Therefore, it can be confirmed that the immunosuppression model through CY is experimentally well implemented.

As shown in Figure 5, the mRNA expression level of IL-1β decreased by 46% in the group treated with CY alone but was significantly recovered (*p* < 0.05) in the HMWF high-treated group. In addition, this trend was also observed in the mRNA expression of IL-2, IL-4, IL-5, IL-12, IFN-γ, and TNF-α. Cytokines play a crucial role in immune response, often working together and helping to synthesize other cytokines [40]. Under these normal conditions, Th1 and Th2 cells are in equilibrium. In order to maintain this balance, it is crucial to control the production of anti- and pro-inflammatory cytokines. Immunomodulation is one of the mechanisms that enhance the body’s defense mechanisms [41], thus increasing its importance in immune-suppressed conditions. According to the above results, HMWF increased mRNA expression levels of Th1 and Th2 cytokines downregulated by CY and contributed to the recovery to CON level. It can be interpreted that HMWF functions more effectively in immuno-suppression model for immune enhancement and equilibrium recovery. In this study, we confirmed that HMWF induces both Th1 and Th2 cytokine secretions. In addition, HMWF induces the secretion of cytokines, restores immunosuppression by CY, and functions as an immunomodulator in the immunosuppressed environment.

In fact, a number of studies have been published on several in vitro immunological efficacies of low molecular weight fraction of *Undaria pinnatifida* and *Hizikia fusiforme* [14,42]. However, studies have shown that high-molecular-weight fucoidan from *Cladosiphon okamuranus* in the peritoneal group effectively changes the CD4+/CD8+ ratio and increases the ratio of cytotoxic T cells in mouse splenocytes compared to that of the low-molecular-weight fucoidan-administered group [43]. It has been reported that the amount of immunoglobulin produced in the spleen lymphocytes of mice treated with high molecular weight fucoidan fractions from sporophyll of *U. pinnatifida* increased [18]. The authors proposed that active elements in the fractionated extract of the brown algae have relatively high-molecular-weight at least above 2,000 MW. Additionally, studies have shown that NK activity is increased by intraperitoneal administration of high-molecular-weight fucoidan of *Undaria pinnatifida* to mice inoculated with P-388 leukemia cells [15]. Other studies have shown that treatment of high-molecular-weight fucoidan of *Undaria pinnatifida* in splenocytes prevents spleen cell necrosis, increases viability, and increases the IFN-γ production [44]. It can be deduced that the difference in the degradation process and absorption pattern of fucoidan in the body also affects the immune activity of fucoidan. Further studies are needed to determine if any structural differences affect the immunoregulatory function of fucoidan, as the mechanisms have not been completely deciphered.

## 3. Materials and Methods

### 3.1. Materials

The brown seaweed *U. Pinnatifida* was cultivated in Wando-gun, Jeonranamdo, Republic of Korea. Chemical reagents, such as sodium nitrate (Duksan, Ansan, Korea), trifluoroacetic acid (Samchun, Pohang, Korea), HCl (Sigma, Poole, United Kingdom), sodium tetraborate decahydrate (Sigma, Poole, United Kingdom), and sulfuric acid (Daejung, Siheung, Korea), were used. The chemical monosaccharide composition of fucoidan was determined using an HPLC system with L-fucose (Sigma, Bratislava, Slovakia), galactose (Sigma, Milan, Italy), mannose (Sigma, Poole, United Kingdom), and D-glucuronic acid (Sigma, St. Louis, MO, USA) as standards. For the cell culture, α-MEM and RPMI-1640 medium were purchased from Gibco Life Technologies (Grand Island, NY, USA). Fetal bovine serum (FBS), streptomycin, and penicillin were purchased from Hyclone, Logan, UT, USA. Concanavalin A (ConA) was purchased by Sigma, St. Louis, MO, USA. The LDH assay kit was provided by Thermo Scientific, Pittsburgh, PA, USA, and the Mouse granzyme B enzyme-linked immunosorbent assay (ELISA) kit was purchased from R&D Systems, Minneapolis, MN, USA.

### 3.2. Extraction and Fractionation of Fucoidan Fraction (FF) into HMWF and LMWF from the Sporophyll of Undaria pinnatifida

A flowchart illustrating the processes of fucoidan extraction and fractionation by MW is shown in Figure 6. Brown seaweed (1 kg) was extracted using 8 kg of 0.83 mM citric acid at 95 °C for 40 min. And then secondary extraction was then implemented to turn it into 4.5 kg of distilled water at 105 °C for 20 min.

Soluble algin in the extract was removed, using calcium chloride, and filtered through diatomite. Subsequently, fucoidan was fractionated according to MW via cross-flow ultrafiltration, separately passing through ultra-filtration membranes with molecular weight cut-offs (MWCOs) of 300 kDa and 10 kDa. HMWF, a fucoidan having an increased average MW of 300 kDa or more, was separated from LMWF, a fucoidan with an average MW of < 300 kDa, using an ultrafiltration membrane (MWCO 300 kDa). The solution was freeze-dried to obtain dried fucoidan.

### 3.3. Chemical Composition of HMWF and LMWF

#### 3.3.1. Total Sugar Content

The phenol-sulfuric acid method [45] was used to measure the total sugars contained in the LMWF and HMWF fractions, and fucose was used as the standard. Standard solution (1.0 mL) was mixed with 1.0 mL 5% (w/w) aqueous phenol solution (Sigma, Poole, United Kingdom) and 5 mL concentrated H_2_SO_4_ (Daejung, Siheung, Korea) in a glass tube. The tube was allowed to stand for at least 25 min in a shaker water bath at room temperature. The sample absorbance was measured at 480 nm (sense; HIDEX, Turku, Finland), and L-fucose (Sigma, Poole, United Kingdom) was used as standard. Serially diluted standards were calculated to obtain the standard curve.

#### 3.3.2. Uronic Acid Content

The uronic acid content of the sample was measured using the sulfamate/m-hydroxydiphenyl assay [46]. The H_2_SO_4_-borate reagent was prepared by dissolving 0.9 g of sodium tetraborate decahydrate (Sigma, Poole, United Kingdom) and 10 mL of distilled water into 90 mL of concentrated sulfuric acid (Daejung, Siheung, Korea). A carbazole reagent was prepared by mixing 0.1 g of carbazole with 100 mL of absolute ethanol in a brown glass bottle. A volume of standard solutions (1.0 mL) was filled with 6 mL of H_2_SO_4_-Borate reagent and cooled in an ice bath. Tubes were then heated for 10 min in a boiling water bath and then cooled in an ice bath. Carbazole reagent (0.2 mL) was added to each tube, then, the tubes were heated for a further 15 min in the boiling water bath, and cooled to room temperature. Absorbance was measured at 525 nm and D-glucuronic acid was used as standard. Serially diluted standards were calculated to obtain the standard curve.

#### 3.3.3. Sulfate Content

A known amount of dried sample (W1, 1 g) was hydrolyzed with 50 mL of 1 N HCl (Sigma, Poole, United Kingdom) for 1 h at 100 °C. To each tube, 10% H_2_O_2_ (25 mL) was added and heated for 5 h in a boiling water bath and filtered. Then, 10 mL of 12% BaCl_2_ was dropped into the tube and heated for a further 2 h in the boiling water bath. After cooling to room temperature, the barium sulfate precipitates were filtered through filter paper and incinerated for 4 h at 600 °C. The filter paper and its contents were dried in an oven until the ash turned white. The white ash (W2) was collected and weighed, and sulfate content was calculated using the equation below:% sulfate = (W2/W1) × 100 × 0.4116(1)

### 3.4. Determination of Monosaccharide Composition of HMWF and LMWF

The monosaccharide content was assessed via an HPLC system comprising a pump (Waters 2695, Milford, MA, USA), Dionex CarboPac™ PA10 column (250.0 mm × 4.0 mm I.D., Dionex corp., Sunnyvale, CA, USA), and refractive index detector (Waters 410, Milford, MA, USA). The sample (100 mg) was treated with 1 mL of 3 M trifluoroacetic acid (Samchun, Pohang, Korea) and 2 mL distilled water at 100 °C for 2 h. After trifluoroacetic acid hydrolysis, the reaction medium was dried using a vacuum concentrator, and distilled water was added to redissolve the sample. The resultant mixture was made up to ~10 mL by adding 18 mM NaOH. The polysaccharide sample (10 μL) was injected into the HPLC system. The column was kept in a 40 °C column oven, and 18 mM NaOH was used as the mobile phase at a flow rate of 0.4 mL/min. The standard was L-fucose, galactose, and mannose. The data were analyzed using the software.

### 3.5. Determination of Molecular Weight by HPGPC

HPGPC was performed with an HPLC system comprising a pump (Waters Alliance 2695, Milford, MA, USA), Waters Ultrahydrogel 500 and 250 columns (250.0 mm × 4.0 mm, Milford, MA, USA), and a refractive index detector (Waters 410, Millipore, Milford, CT, USA).

HMWF and LMWF (30 mg/10 mL) were prepared and filtered through 0.45 μm syringe filters into HPLC vials. The sample (100 μL) was injected into the HPLC system. The column was kept in a 40 °C column oven and 0.2 M NaNO_3_ was used as the mobile phase at a flow rate of 1 mL/min. The standard was pullulan kit (MW 1300 to 1,300,000 Da, Shodex, Japan) and the MWs of the extracted fucoidan were determined using the HPLC software.

### 3.6. In Vitro Assay

#### 3.6.1. Cell Culture and Proliferation Assay

A human natural killer cell line, NK-92MI cell (ATCC, Manassas, VA, USA) was maintained in an α-MEM medium that was supplemented with 100 μg/mL streptomycin, 100 U/mL penicillin, 10% FBS, 10% horse serum, 0.2 mM inositol, 0.02 mM folic acid, and 0.1 mM β-mercaptoethanol. Cell proliferation was assessed by using EZ-Cytox cell viability assay kit (Daeill Lab Service Co., Seoul, Korea). NK cells were plated in 96-well plates to a concentration of 2 × 10^5^ cells/well and co-incubated with the samples at various concentrations (125–1000 μg/mL).

#### 3.6.2. NK Cell Cytotoxic Activity Assay

NK-92MI cells were treated with 1,000 μg/mL LMWF and HMWF for 24 h. Activated NK cells act as effector cells against target cells, such as YAC-1 tumor cells, HeLa cells, and K562 cells. Target YAC-1 cells (4 × 10^4^ cells/well) with the E: T ratios, 2:1, 5:1, and 12.5:1 were incubated in a 5% CO_2_ incubator at 37 °C in a 96-well plate and the experiments were carried out three times. After 4 h of culture, NK activity was measured using the Pierce ™ LDH Cytotoxicity Assay Kit.

### 3.7. In Vivo Assay

#### 3.7.1. Treatment and Experimental Design

The mice were obtained from Orient Bio Inc. (Gapyung, Korea). 10-week-old BALB/c mice, weighing 19-20 g, were refined for one week in an experimental laboratory resource center. The mice were randomly divided into 4 groups of 6 mice each and sub-divided into two groups, a control group, and a CY-treated only group and two groups administered two different HMWF concentrations (25 and 75 mg/kg b.w.). CY is a type of immunosuppressant with side effects, such as a decrease in the lineage of blood cells, and its functional products include cytokines [5]. Therefore, it can be used as an inhibitor of immune responses in vitro and in vivo, as in this experiment [47,48]. CY was used to implement the immunodeficiency model in this experiment. The HMWF-treated groups were orally administered at the corresponding concentration for about 28 d, and the control and the CY groups received only oral administration of the same amount of saline. CY (100 mg/kg b.w.) was injected intravenously 7 d before the end of the experiment in mice of all groups except the control group. Animals were sacrificed 24 h after the last oral dose. All subjects gave their informed consent for inclusion before they participated in the study. The study was conducted by the Declaration of Helsinki, and the protocol was approved by the Korea University Institutional Animal Care & Use Committee (KUIACUC-2018-64).

#### 3.7.2. Splenic NK Cell Cytotoxic Activity Assay

Splenic NK cell activity was determined by a modified previously published method [49]. Splenocytes were harvested immediately after sacrifice from 6 mice per group-administered orally for 28 d. Separated splenocytes were counted using trypan blue reagent. Splenocytes act as effector cells against YAC-1 tumor cells (target cells). The effector-to-target cell (E:T) ratios were 4:1, 10:1, and 25:1 for cells incubated in 96-well plate at 37 °C in a CO_2_ incubator. After 4 h of culture, NK activity is measured using the Pierce™ LDH Cytotoxicity Assay Kit. The experiment was carried out three times in the same manner.

#### 3.7.3. Granzyme B Secretion and Quantification by ELISA

Cells were cultured in a 96-well plate at 37 °C in a 5% CO_2_ incubator at E:T ratios 4:1, 10:1, and 25:1, where splenocytes were the effector cells and YAC-1, the target cells. The supernatant was obtained from splenocytes by centrifugation at 600× *g* for 5 min. The content of granzyme B in the supernatant was quantified according to the manufacturer’s instructions using mouse ELISA kits (R & D systems, Minneapolis, MN, USA).

#### 3.7.4. Quantitative Reverse Transcription-Polymerase Chain Reaction Assay

The quantitative reverse transcription-polymerase chain reaction (qRT-PCR) assay was performed as previously described [50] to measure cytokine expression levels of spleens harvested from each group. Total RNA was extracted through RNAiso plus reagent (Takara, Otsu, Shiga, Japan). cDNA was synthesized using LeGene Premium Express 1st strand cDNA synthesis system. Gene expression levels were measured by quantitative reverse transcription amplification using the iQ5 real-time PCR detection system (Bio-Rad, Hercules, CA, USA) and SYBR green master mix (Elpis, Daejeon, Korea). The mouse qTR-PCR primer sequences were shown in Appendix A. Rn18s was calibrated using the housekeeping gene and the data were measured using the 2-ΔΔCt method [51].

### 3.8. Statistical Analysis

All statistical analyses were performed using the SAS version 9.4 (SAS institute, Cary, NC, USA). Quantified data were expressed as mean ± SD. All statistical comparisons were made by one-way ANOVA followed by Tukey and Duncan’s multiple range test. The differences between groups with *p*-values < 0.05 were considered statistically significant.

## Figures and Tables

**Figure 1 marinedrugs-17-00447-f001:**
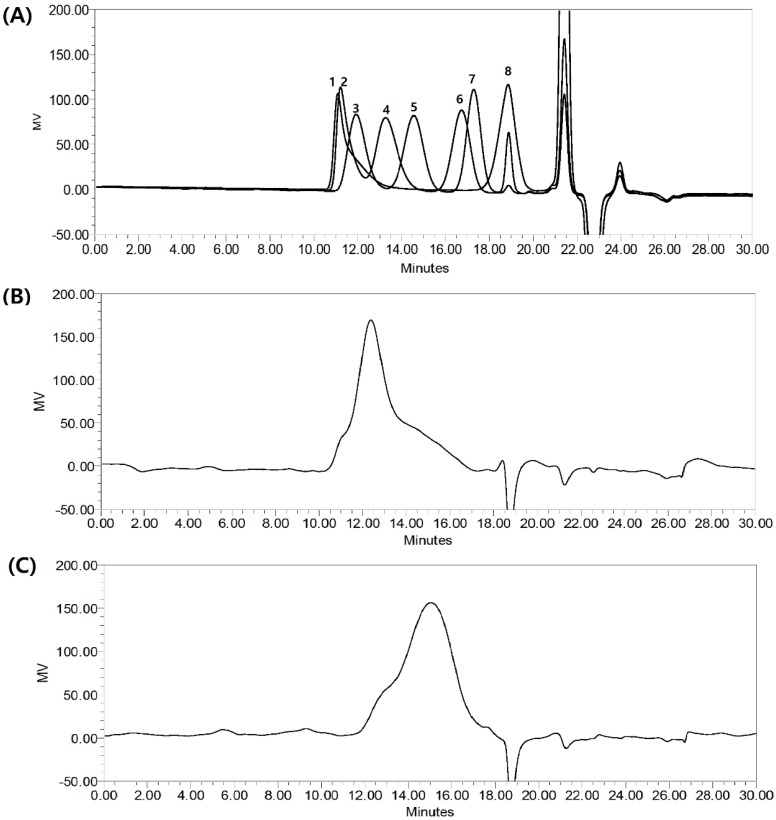
The molecular weight of HMWF and LMWF. High-perormance gel permeation chromatography analysis of fucoidan fractions (HMWF, LMWF), through a cross-flow ultrafiltration system (molecular weight cut-off of 300 kDa). (**A**) pullulan standard: 1. 1,330 kDa, 2. 642 kDa, 3. 344 kDa, 4. 47 kDa, 5. 10.7 kDa, 6. 9.6 kDa, 7. 6.1 kDa, 8. 1.32 kDa. (**B**) HMWF, (**C**) LMWF.

**Figure 2 marinedrugs-17-00447-f002:**
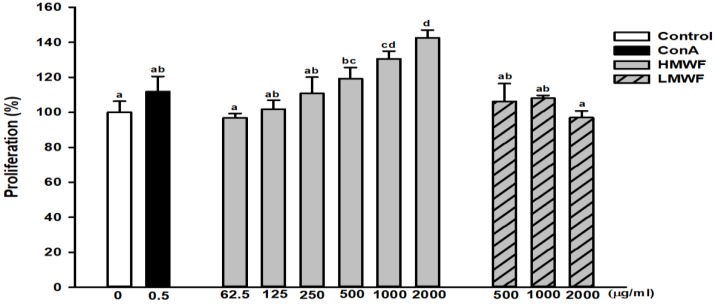
Effects of HMWF and LMWF on natural killer (NK) cell, NK-92MI, proliferation. The NK cells were incubated with the indicated concentrations of concanavalin A (ConA), HMWF or LMWF for 24 h. Values are presented as mean ± standard deviation (*n* = 3). Different letters (a–d) indicate significant differences from the CON group at *p* < 0.05 by Tukey’s post-hoc test.

**Figure 3 marinedrugs-17-00447-f003:**
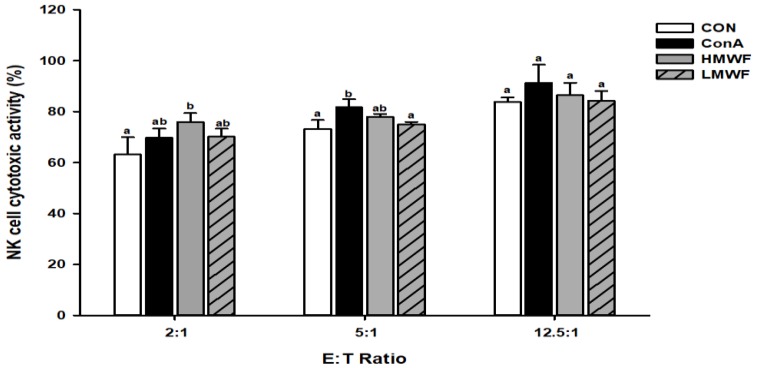
Effects of HWMF and LMWF on NK cell cytotoxic activity in NK-92MI cells. NK cells were incubated with the indicated concentrations of HMWF and LMWF for 24 h. After incubation, NK cells were co-incubated with YAC-1 cells, as target cells, in effector-to-target ratios of 2:1, 10:1, and 50:1 for 4 h. NK cytotoxic activity was measured by lactate dehydrogenase (LDH) assay. Values are presented as mean ± standard deviation (*n* = 3). Different letters indicate significant differences from the CON group at *p* < 0.05 by Tukey’s post-hoc test. CON: control; ConA: concanavalin A; HMWF: high molecular weight fucoidan; LMWF: low molecular weight fucoidan.

**Figure 4 marinedrugs-17-00447-f004:**
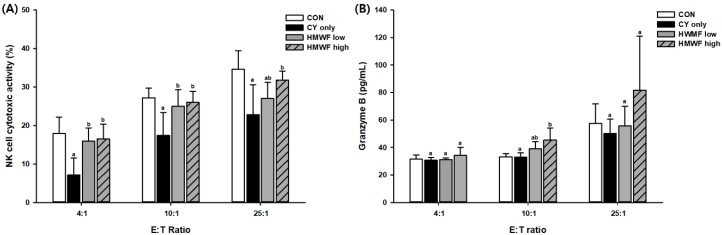
Effect of HMWF on NK cell cytotoxic activity and granzyme B release in cyclophosphamide (CY)-treated mice. Immunosuppression was induced by injection of CY (100 mg/kg body weight (b.w.)/d, i.p.) on day 21. The control (CON) group and CY-treated only (CY only) group was orally administered with vehicles alone from day 1 to day 28, while the HWMF-low treated group (HMWF low), and HMWF- high treated group (HMWF high), were administered HWMF at a dosage of 25 and 75 mg/kg b.w., respectively. The CY only group, HMWF-, and LMWF-treated groups were injected with CY on day 21. Splenocytes were incubated with YAC-1 cells for 4 h. (**A**) NK cell cytotoxic activity was measured by LDH assay. (**B**) Granzyme B release was measured by enzyme-linked immunosorbent assay. Values are presented as mean ± standard deviation (*n* = 6). Different letters indicate significant differences from the CY only group at *p* < 0.05 by a Duncan test.

**Figure 5 marinedrugs-17-00447-f005:**
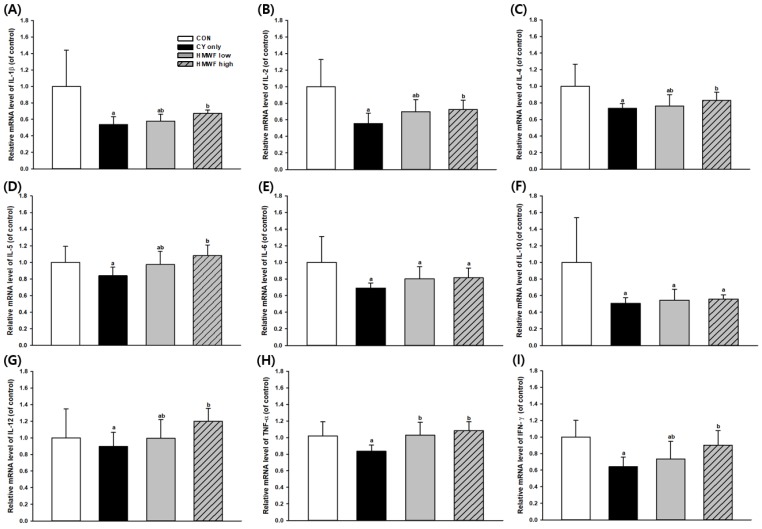
Effects of HWMF on cytokine mRNA expression in CY-treated mice. Immunosuppression was induced by injection of CY (100 mg/kg/d, i.p.) on day 21. (**A**–**I**) Graphic representation of mRNA expression levels of cytokines, interleukin (IL)-1β, IL-2, IL-4, IL-5, IL-6, IL-10, IL-12, tumor necrosis factor (TNF)-α, and interferon (IFN)-γ, respectively. The CY-treated only (CY only) group was orally administered alone from day 1 to day 28, while HMWF-treated groups, HWMF low group, and HMWF high group, were administered HWMF at a dosage of 25 and 75 mg/kg b.w., respectively. Values are presented as mean ± standard deviation (*n* = 6). Different letters indicate significant differences from the CY only group at *p* < 0.05 by a Duncan test.

**Figure 6 marinedrugs-17-00447-f006:**
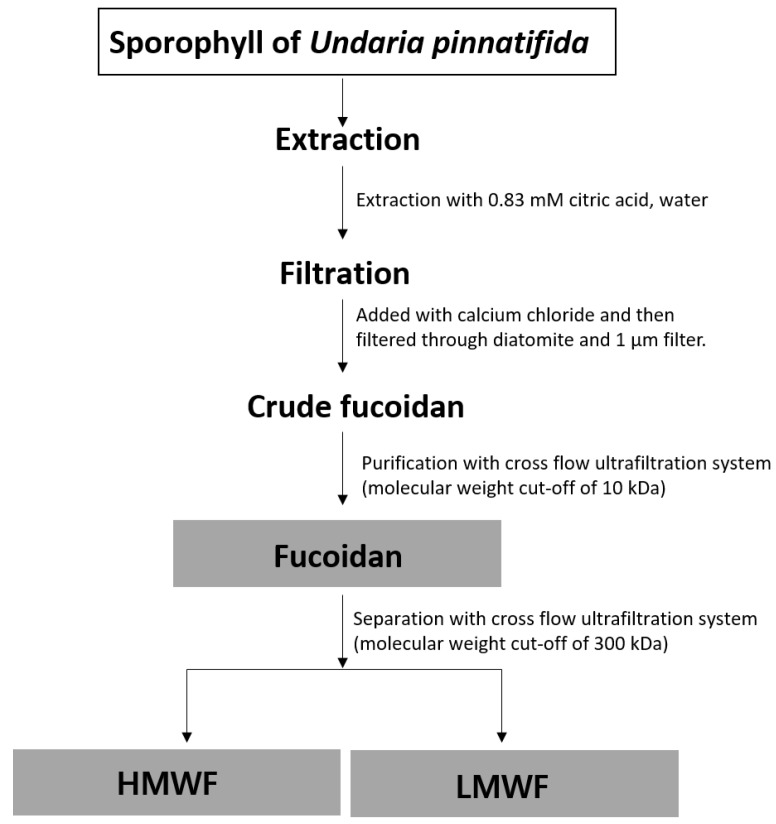
Flowchart illustrating the extraction and fractionation of HMWF and LMWF from the sporophyll of *Undaria pinnatifida.*

**Table 1 marinedrugs-17-00447-t001:** The chemical composition of HMWF ^1^ and LMWF ^2^. HMWF, a fucoidan with an average molecular weight (MW) of 300 kDa or more was separated from LMWF, a fucoidan with an average MW < 300 kDa, using an ultrafiltration membrane having a MW cut-off of 300 kDa.

Sample	Carbohydrate (%)	Sulfate (%)	Uronic Acid (%)	Monosaccharide Composition		
				Fucose (%)	Galactose (%)	Mannose (%)
HMWF ^1^	72.0 ± 3.9	30.9 ± 2.7	10.9 ± 2.8	21.0 ± 2.3	23.0 ± 0.8	0.9 ± 0.5
LMWF ^2^	73.8 ± 3.3	28.8 ± 2.7	10.3 ± 2.5	21.5 ± 1.5	16.7 ± 2.7	0.75 ± 0.2

^1^ HMWF: high molecular weight fucoidan; ^2^ LMWF: low molecular weight fucoidan.

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
