# Peer review of "Characterization and Immunomodulatory Effects of High Molecular Weight Fucoidan Fraction from the Sporophyll of Undaria pinnatifida in Cyclophosphamide-Induced Immunosuppressed Mice"

_marinedrugs, 2019, doi:10.3390/md17080447_

Reviewer 1 Report

The manuscript entitled “Characterization and immunomodulatory effects of high molecular weight fucoidan fraction from the sporophyll of Undaria pinnatifida in cyclophosphamide-induced immunosuppressed mice”. No doubt the subject presented is interesting and it deserves further studies.

Manuscript presented is well written and organized.

Molecular mass, monosaccharide composition and sulphate contents of fucoidans that used in experiments was establish.  Methods used is adequate to aim of work.

In my opinion, disadvantageous feature of the work is a part on the structure investigation of fucoidans used.  It is well known that fucoidans isolated from Undaria pinnatifida have very different structure.  Information about sulphate positions, type of glycosidic bonds in polysaccharides is essential, but authors not include these data in manusctipt. Authors should include at least NMR spectra of HMWF and LMWF. Absence of these data greatly reduces the value of the presented work.

Investigation of immunomodulating activity was done on high level and beyond any doubt.

Author Response

Reviewer 1:

Comments to the Author

The manuscript entitled “Characterization and immunomodulatory effects of high molecular weight fucoidan fraction from the sporophyll of Undaria pinnatifida in cyclophosphamide-induced immunosuppressed mice”. No doubt the subject presented is interesting and it deserves further studies.

Manuscript presented is well written and organized.

Molecular mass, monosaccharide composition and sulphate contents of fucoidans that used in experiments was establish.  Methods used is adequate to aim of work.

Comments:

1) In my opinion, disadvantageous feature of the work is a part on the structure investigation of fucoidans used.  It is well known that fucoidans isolated from Undaria pinnatifida have very different structure. Information about sulphate positions, type of glycosidic bonds in polysaccharides is essential, but authors not include these data in manuscript. Authors should include at least NMR spectra of HMWF and LMWF

Answer:

First of all, we’d like to express our gratitude to the reviewer for the careful and critical reading of our manuscript.

Unfortunately, we could not add the structural characterization data you mentioned in our manuscript due to the predetermined experiment design.

In industry related activities in South Korea, low molecular weight fucoidan material (MW<300 kDa) from Undaria pinnatifida as raw material for cosmetics has been utilized, whereas high molecular weight fucoidan (MW ³ 300 kDa) has been used less. This is a background of this study.

In fact, when compared to low-molecular-weight fucoidan, the in vivo research on the immunomodulatory effect and biological function of high-molecular-weight fucoidan is relatively limited.

Please understand that this study focuses on in vivo immune activity of this molecular weight of high molecular weight fucoidan. We provied approprite descriptions in lines (L) 67-74, page (P) 2 as follow:

“Using analytical methods the links between fucoidan monosaccharides were discovered to be 1-3 linked fucose and 1-3, 1-4, and 1-6 linked galactose in the glycosidic bonds, and sulfate substitution sites are primarily at C2- or C4-positions of residues of fucose, and C3- or C6-positions of residues of galactose [21]. The structure of HMWF and LMWF fractionated according to MW via cross-flow ultrafiltration from U. pinnatifida was not determined in our study. Given the distinction between HMWF and LMWF in the content of galactose residues and sulfate content, it can be seen that these two fucoidans appear to differ in structure. Future study on the structural characterization of HMWF and LMWF is required.”

2) Investigation of immunomodulating activity was done on high level and beyond any doubt.

Answer:

We express our gratitude for your comment.

Reviewer 2 Report

The study reports interesting results on the immustimulating effects of high molecular weight fucoidan compared to low molecular weight fucoidan, based on both in vitro and in vivo data. The authors conclude that high molecular weight fucoidan induces the secretion of cytokines, restores immunosuppression by CY, and functions as an immunomodulator in the immunosuppressed environment. 

The paper is overall well written, clear and concise. Nevertheless, I suggest some essential revisions:

Figure captions. Please indicate the significance of a, ab, bc, cd and d.

The controls denomination is not clear between in vitro and in vivo experiments. A negative control is supposed to have no effect, yet you use the name NC for mice injected with CY alone. It should be named positive control.

The description of Figure 4B doesn't match the results. Please explain why granzyme B release is not lower in NC group than in control group whatever the E:T ratio. Is granzyme B a good marker for CY side effects? 

The discussion on why HMWF is more efficient than LMWF to promote immunostimulation is lacking. Authors should at least raise hypothesis.

Materials and methods. Could the authors specify if the fucoidans were incubated with the cells before seeding in microplates or after? If the fucoidans were incubated simultaneously with the cells, then the authors studied the effects of fucoidans on cell adhesion rather than cell proliferation. In proliferation studies, fucoidans should be incubated with the cells once they have attached to the microplates. 

3.6.2. Correct the word thrice.

3.7.1. and 3.7.2. Authors used the word rats instead of mice

Author Response

Reviewer 2:

Comments to the Author

The study reports interesting results on the immustimulating effects of high molecular weight fucoidan compared to low molecular weight fucoidan, based on both in vitro and in vivo data. The authors conclude that high molecular weight fucoidan induces the secretion of cytokines, restores immunosuppression by CY, and functions as an immunomodulator in the immunosuppressed environment. 

The paper is overall well written, clear and concise. Nevertheless, I suggest some essential revisions:

1) Figure captions. Please indicate the significance of a, ab, bc, cd, and d.

Answer:

First of all, we’d like to express our gratitude to the reviewer for the careful and critical reading of our manuscript.

We corrected the caption by adding an exact description of the statistical significance mentioned. Please see the captions for Figs. 2 to 5.

2) The controls denomination is not clear between in vitro and in vivo experiments. A negative control is supposed to have no effect, yet you use the name NC for mice injected with CY alone. It should be named positive control.

Answer:

We have found that the term of negative control for the mice injected with CY alone can cause confusion for the readers. The group named as a negative control in this paper is a group to examine the recovery effect of fucoidan samples in order to confirm that the immunocompromised animal model is well implemented by treating only cyclophosphamide without treatment of fucoidan sample.   

Therefore, we changed the ‘negative control (NC)’ to 'CY-treated only group (CY only)' in lines (L) 144, 145, 151, 155, 164 pages (P) 5; L180, 183, 187, P6; L230, 234, P7; L364, P10.

3) The description of Figure 4B doesn't match the results. Please explain why granzyme B release is not lower in NC group than in control group whatever the E:T ratio. Is granzyme B a good marker for CY side effects? 

Answer:

According to the previously published studies, "Granzyme B genes are mainly down-regulated upon treatment with the cytotoxic activity of the immune cells mainly mediated by NK cells"[1]. In addition, the treatment of CY is involved in the reduction of granzyme B and perforin activity which can lead to apoptosis[2]. Therefore, granzyme B can be a marker to see the side effect of CY.

Activated NK cells also secrete apoptosis-inducing granules, one of which is granzyme B, showing that the inhibition by CY is restored by the sample treatment.

Based on our data, granzyme B release of the CON group is 57.48 pg mL at E:T ratio of 25:1, and this is lower value compared to the value of 50.12 pg/mL in the CY only group. However, since the splenocyte was obtained by grinding the spleen of the harvested animal, and the amount of released granzyme B in the supernatant was measured, the deviation between the animals was large and the significant difference between two groups was not observed.

[1] El-Serafi, I., Abedi-Valugerdi, M., Potácová, Z., Afsharian, P., Mattsson, J., Moshfegh, A., & Hassan, M. (2014). Cyclophosphamide alters the gene expression profile in patients treated with high doses prior to stem cell transplantation. PloS one9(1), e86619

2 Lee, J., & Lim, K. T. (2012). SJSZ glycoprotein (38 kDa) modulates expression of IL-2, IL-12, and IFN-γ in cyclophosphamide-induced Balb/c. Inflammation Research61(12), 1319-1328.

4) The discussion on why HMWF is more efficient than LMWF to promote immunostimulation is lacking. Authors should at least raise hypothesis.

Answer:

We have added descriptions in L107-110, P4 as follows:

“NK cells engage in target cell lysis and activate cytokine secretion [29] as well. Therefore, if treated at the same concentration, a marked rise in the number of NK cells in the HMWF-treated group means that it may be more effective in immunomodulation.”

We have also found an article showing that high molecular weight fucoidan can function more effectively as an immunomodulator than low molecular weight fucoidan. We added their result in L239-243, P7 as follows:

“It has been reported that the amount of immunoglobulin produced in the spleen lymphocytes of mice treated with high-molecular-weight fucoidan fractions from sporophyll of U. pinnatifida increased [45]. The authors proposed that active elements in the fractionated extract of the brown algae have relatively high molecular weight at least above 2,000 MW.”

 5) Materials and methods. Could the authors specify if the fucoidans were incubated with the cells before seeding in microplates or after? If the fucoidans were incubated simultaneously with the cells, then the authors studied the effects of fucoidans on cell adhesion rather than cell proliferation. In proliferation studies, fucoidans should be incubated with the cells once they have attached to the microplates. 

Answer:

We apologize for any inconvenience caused by the lack of a detailed description of the experiment and method. We have tried to reduce the confusion by changing the contents of the experiment in L 343-346, P11 as follows:

“Cell proliferation was assessed by using EZ-Cytox cell viability assay kit (Daeill Lab Service Co., Seoul, Korea). NK cells were plated in 96-well plates to a concentration of 2×105 cells/well and co-incubated with the samples at various concentrations (125 – 1000 μg/mL).”

Compared to the MTT assay, there is a difference between the experimental method and the principle.

First, since NK-92MI cells are floating cells, the cells do not need an attachment process. We co-treated with fucoidan in the experiment. After the treatment time, the EZ-CYTOX reagent is placed in each well and the absorbance is measured at 450 nm. Because it measures the mitochondrial activity of living cells in the process, it is often used as one of the methods for measuring cell viability. Therefore, when proliferation occurs in floating NK cells, the number of existing and increased cells can be measured through EZ-Cytox reagent.

6) 3.6.2. Correct the word thrice.

Answer:

We have revised ‘thrice’ to 'three times' in L356, P10.

7) 3.7.1. and 3.7.2. Authors used the word rats instead of mice

Answer:

We corrected all the parts that were incorrectly described as 'rats' to 'mice' in L 363, 371, 375, P 10.

We used Balb/c mice as an experimental animal. We are so sorry for our mistake.